# Rehabilitation Training after Spinal Cord Injury Affects Brain Structure and Function: From Mechanisms to Methods

**DOI:** 10.3390/biomedicines12010041

**Published:** 2023-12-22

**Authors:** Le-Wei He, Xiao-Jun Guo, Can Zhao, Jia-Sheng Rao

**Affiliations:** 1Beijing Key Laboratory for Biomaterials and Neural Regeneration, Beijing Advanced Innovation Center for Biomedical Engineering, School of Biological Science and Medical Engineering, Beihang University, Beijing 100191, China; hlw0118@buaa.edu.cn (L.-W.H.); guoxiaojun0513@163.com (X.-J.G.); 2Institute of Rehabilitation Engineering, China Rehabilitation Science Institute, Beijing 100068, China

**Keywords:** spinal cord injury, exercise, brain reorganization, rehabilitation training

## Abstract

Spinal cord injury (SCI) is a serious neurological insult that disrupts the ascending and descending neural pathways between the peripheral nerves and the brain, leading to not only functional deficits in the injured area and below the level of the lesion but also morphological, structural, and functional reorganization of the brain. These changes introduce new challenges and uncertainties into the treatment of SCI. Rehabilitation training, a clinical intervention designed to promote functional recovery after spinal cord and brain injuries, has been reported to promote activation and functional reorganization of the cerebral cortex through multiple physiological mechanisms. In this review, we evaluate the potential mechanisms of exercise that affect the brain structure and function, as well as the rehabilitation training process for the brain after SCI. Additionally, we compare and discuss the principles, effects, and future directions of several rehabilitation training methods that facilitate cerebral cortex activation and recovery after SCI. Understanding the regulatory role of rehabilitation training at the supraspinal center is of great significance for clinicians to develop SCI treatment strategies and optimize rehabilitation plans.

## 1. Introduction

According to the data of the World Health Organization (WHO), 250,000–500,000 people worldwide suffer from spinal cord injury (SCI) every year [1]. SCI typically entails structural damage to the spinal cord, leading to disruption in the transmission of sensory and motor information, thereby causing dysfunction below the affected region. This debilitating condition significantly impacts patients’ daily lives and imposes substantial economic burdens. Despite the grave consequences of SCI, a viable clinical treatment approach remains elusive. The repair of the central nervous system (CNS) after SCI has been a focal point of biomedical research.

The nervous system is an essential component of the body. When SCI occurs, it cuts off the crucial information flow between the body and the brain, thereby affecting not only the areas below the level of injury but also the structure and function of the brain [2,3]. The CNS has exceptional plasticity, and this plasticity has the potential to compensate for damage. When the components of the brain are disrupted, the cerebral cortex undergoes neural network reorganization [4]. This process involves changes in neurotrophin secretion, cytokine content, biochemical composition, and nerve cell morphology, all of which contribute to the creation of appropriate new synapses and neural circuits, leading to the reorganization and compensation of the brain [2]. The ability of the brain to reorganize and compensate is crucial to the improvement of brain structure and function after SCI [5].

Rehabilitation training, a type of physical therapy that utilizes physical factors to stimulate adaptive changes in local neural circuits, is used widely in the clinical treatment of SCI to improve the dysfunction caused by SCI [6]. Previous research has shown that rehabilitation training can promote cell activation and functional remodeling in the cerebral cortex [7]. This manuscript will review the effects of rehabilitation training on brain structure and function after SCI, as well as examine how different rehabilitation methods affect brain reorganization.

## 2. Effect of Rehabilitation Training on Brain Structure and Function

### 2.1. Physiological Mechanism of Exercise and Brain Structure and Function

Exercise is beneficial for the development of muscle and physical fitness, and it is important for the activation and promotion of neuronal connections [8]. Kobilo et al. [9] found that exercise can increase the volume of the hippocampus and blood flow to this part of the brain through studies of mouse models. In different mouse models, exercise increases synaptic plasticity and induces morphological changes in dendrites [10,11]. When part of the brain structure is damaged, the use of rehabilitation training is currently considered to promote functional recovery. In addition to its effects on the local area, rehabilitation training can also affect neural connections across different brain regions. Batouli et al. [12] found that physical activity can change the brain neural network, which accounts for 82% of the total gray matter volume. Another study involving functional magnetic resonance imaging (fMRI) in patients with cervical SCI showed that the degree of activation of the cerebral cortex related to exercise is closely related to the degree of functional improvement after exercise [13].

Recent studies have shown that skeletal muscle and the brain are connected through a muscle–brain endocrine loop in the human body [14]. Skeletal muscle, acting as a secretory organ, can produce and release hundreds of myokines that communicate with other organs during muscle cell proliferation, differentiation, or muscle contraction [15,16]. Myokines can be transmitted to the liver, pancreas, bones, blood vessels, and other organs and participate in the direct regulation of brain structure and function [17,18]. In addition, skeletal muscle can participate in neural feedback to the brain through the peripheral–central nervous pathway and regulate brain functional activity and network integration through the ascending sensory information flow [14].

### 2.2. Effect of Rehabilitation Training on the Brain after SCI

#### 2.2.1. Rehabilitation Training Regulates the Secretion of Neurotrophic Factors in the Brain

Brain-derived neurotrophic factor (BDNF) is an essential regulator of neuronal development, synaptic transmission, and cell and synaptic plasticity. It promotes the survival, differentiation, and growth of neurons and glial cells [19]. Rehabilitation training can stimulate the release of BDNF, thereby affecting brain structure and function. When mice engage in voluntary exercise, the level of BDNF in the hippocampus significantly increases compared to mice that do not exercise [20]. This exercise-dependent cell proliferation in the dentate gyrus of the hippocampus plays a role in some forms of memory information and maintenance. Another study showed that rehabilitation training can increase BDNF levels in the brain and spinal cord of mice, alleviate axonal demyelination after SCI, and reduce the size of the injured area [21]. Several clinical studies have also shown that BDNF levels are associated with rehabilitation training; for example, 9 weeks of aerobic and pilates training can increase human serum BDNF [22], while 3 months of aerobic exercise training increased the hippocampal volume of patients with schizophrenia by 16% [23]. A half-year resistance and endurance training can increase the serum BDNF concentration of patients with multiple sclerosis by 13.9% [24].

Aside from directly affecting BDNF concentration, rehabilitation training can regulate the motor–brain pathway related to BDNF [25]. During rehabilitation training, cathepsin B, which are myokines released by skeletal muscle, can affect neurogenesis, learning, and memory. Moon et al. [26] used AMPK agonists to treat myotubes and simulated the effect of exercise in vitro. Results show that AMPK can induce skeletal muscle cells to secrete cathepsin B. Moreover, running can increase the levels of cathepsin B in the gastrocnemius muscle of mice, and the levels of actin and cathepsin B in the plasma of rhesus monkeys and humans. Cathepsin B crosses the blood–brain barrier and enters the CNS to upregulate BDNF concentration [26]. Actin can participate in the cAMP-dependent regulation of BDNF transcription and is the main driving force of dendritic spine remodeling and sustains synaptic plasticity [27,28].

Rehabilitation training can regulate BDNF via the “PGC1α-FNDC5-irisin” pathway. PGC1α has a central role in mediating many of the metabolic effects of exercise locally within the muscle. During rehabilitation training, PGC1α expression in muscle increases and stimulates an increase in the expression of FNDC5 [29]. FNDC5 is a membrane protein that is cleaved and secreted into the circulation as the myokine irisin [30], causing an increase in irisin levels in the blood. And irisin passes through the blood–brain barrier, upregulating BDNF level in the brain [29].

In addition, rehabilitation training can upregulate BDNF expression by stimulating an increase in blood β-hydroxybutyric acid levels [31].

#### 2.2.2. Rehabilitation Training Regulates Cytokines and Biochemical Changes in the Brain

Rehabilitation training affects the level of cytokines in the CNS and tends to create an immunosuppressive environment in the CNS to combat chronic brain inflammation induced by SCI. Rehabilitation training can significantly increase the level of cytokine IL-6 and produce anti-inflammatory effects by inhibiting the increase in circulating TNF levels induced by endotoxin [32]. Additionally, rehabilitation training can reduce the levels of proinflammatory cytokines such as IL-1β and IL-17 and increase the levels of anti-inflammatory cytokines such as IL-10 and transforming growth factor (TGF)-β [33,34,35]. Nichol et al. [36] found that 3 weeks of voluntary wheel training can effectively reduce the levels of proinflammatory cytokines IL-1β and TNF-α in the hippocampus of rats and make the level of inflammatory cytokines IFN-γ return to normal, which can suppress inflammatory response in the brain [37].

Rehabilitation training can also cause changes in biochemical levels in some parts of the brain. In rats undergoing treadmill rehabilitation training, the nucleolar area of motor neurons increased, and the protein synthesis related to plasticity in the cerebral cortex increased significantly [38]. Rehabilitation training can also promote the ability of motor neurons to transport positive/reverse axonal proteins, such as the highly selective transport of the synaptic protein SNAP25, thus improving the overall adaptive capacity of motor units [39].

Rehabilitation training can also promote the functional regulation of brain mitochondrial machinery. The density and function of mitochondria in the brain of adult mice increase after 4 weeks of voluntary wheel exercise [40]. Another study has shown that endurance training can induce upregulation of PGC-1α and SIRT1, activation of AMPK, decrease in p53 acetylation, and increase in mitochondrial respiratory complex content in the rodent brain [41]. These factors may have an impact on the brain, regulate brain plasticity mechanisms, and induce neuroprotective pathways. PGC-1α can up-regulate BDNF in the brain through the “PGC1α-FNDC5-irisin” pathway [29]. SIRT1 is an NAD+-dependent deacetylase. Evidence indicates that the SIRT1 1 pathway regulates cell survival and rescues proteins in several nervous system diseases [42]. SIRT1 mediates AMPK action on PGC-1α transcriptional activity and the expression of genes involved in mitochondrial activity [43]. Simultaneously, the decrease in p53 acetylation can promote the expression of SIRT1 [44]. The CNS requires a large amount of energy to conduct nerve impulses, and the enhancement of the density and function of mitochondria in the brain is beneficial to neuronal activity [45]. At the same time, the ability of mitochondria to regulate intracellular Ca2+ plays an important role in synaptic maintenance [46]. These mechanisms are beneficial to protect residual nerve tissue and function after SCI and reduce the negative effects of injury.

#### 2.2.3. Effect of Rehabilitation Training on Nerve Cells

Oligodendrocytes, astrocytes, microglia, and other glial cells are widely distributed in the CNS [47]. These cells can connect and support various neural components and play an important role in the structure and function of the brain. SCI can lead to apoptosis or proliferation of nerve cells, thus affecting the original brain structure and function [48]. Therefore, normalizing the type and number of nerve cells in the brain needs to maintain the structure and function of the brain through rehabilitation training.

##### Oligodendrocyte

Oligodendrocytes are a major type of glial cells in the CNS, which can produce lipid-rich myelin around the axons of the CNS, thereby insulating axonal segments to increase axonal conduction velocity and provide nutritional support for axons by secreting metabolic factors such as lactic acid [49]. Oligodendrocyte progenitor cells (OPCs) originate from neural stem cells in the ventricular region and are evenly distributed in the CNS. In response to local proliferation and differentiation signals at their final location in the CNS, OPCs differentiate into mature oligodendrocytes [50].

Previous studies have shown that, compared with mice without exercise, the number of proliferating OPCs in mice undergoing exercise on a running wheel was significantly higher [51]. Kim et al. [52] found that rehabilitation training promoted the redifferentiation of oligodendrocytes and remyelination of axons by immunolabeling MBP- and CNPase-positive cells in the hippocampus of mice after exercise. Even without CNS injury, exercise is beneficial for oligodendrocyte formation in mice; for instance, 7 days of wheel exercise increased the proliferation of immature oligodendrocytes in the spinal cord [53]. Although exercise can increase the number of oligodendrocytes, the mechanism by which exercise promotes oligodendrocyte regeneration in diseased microenvironments remains to be further clarified.

##### Astrocytes

Astrocytes are a group of highly heterogeneous glial cells in the CNS. The structural diversity of astrocytes enables astrocytes to perform multiple functions [54]. Astrocytes play a crucial role in synaptic transmission and information processing, as well as the function of neural circuits [54]. Additionally, astrocytes serve important molecular functions in the formation, maintenance, and pruning of synapses, and they can regulate the homeostasis of the CNS by modulating the permeability of the blood–brain barrier and energy homeostasis [55].

After damage to the structure and function of the nervous system, astrocytes can undergo reactive hyperplasia with significant changes in gene expression, cell morphology, and function [56]. This process results in the formation of glial scars, which can promote demyelination and neurodegeneration and facilitate the penetration of peripheral immune cells into the brain through the blood–brain barrier [57].

Although SCI does not directly destroy brain structure, it affects cortical neurons through secondary degeneration; the application of rehabilitation training can help maintain neuronal function and synaptic plasticity by inhibiting astrocyte changes in this process. This is supported by several studies on brain diseases, including those by Zhang et al. [58], who found that long-term running reduced astrocyte proliferation and increased neuronal density in Alzheimer’s mice, and Lee et al. [59], who performed treadmill rehabilitation exercises beginning within 24 h after cerebral ischemia injury and found that this process reduced astrocyte formation and promoted the functional recovery of rats.

Exercise can increase shear stress in endothelial cells, regulate the release of soluble factors such as CO, and subsequently regulate Ca2+ in astrocytes. This may be one of the potential mechanisms by which exercise regulates astrocyte injury response [60].

##### Microglia

Microglia are essential glial cells and the first line of immune defense within the CNS. The microglial population is heterogenous and can be simply classified into pro-inflammatory M1 and anti-inflammatory M2 phenotypes [61]. M1 microglia present antigens to T cells and release proinflammatory cytokines, which destroy oligodendrocytes and myelin, resulting in axonal demyelination. In contrast, M2 microglia participate in reducing inflammation by reducing glutamate excitotoxicity and increasing the release of anti-inflammatory cytokines [61].

Previous studies have shown that rehabilitation training can promote the activation of M2-like microglia in mice, which is beneficial for the protection of the CNS [62]. Subsequent studies have further revealed that rehabilitation training can reduce the number of M1-like microglia expressed in the spinal cord during CNS injury, thereby reducing the degree of inflammatory response in the body [63]. In animal models of demyelination, rehabilitation exercise may increase the proportion of M2 phenotypic microglia in the focal area and alleviate the progress of demyelinating lesions [64].

Therefore, rehabilitation training can reduce inflammation in the brain by activating M2 microglia or inhibiting M1 microglia, thus maintaining the stability of structure and function.

#### 2.2.4. Rehabilitation Training Promotes Cortical Remodeling

The function and structure of the spinal cord and brain are closely related. The dysfunction of conduction after SCI affects the structure and function of the brain, leading to the reorganization of the cerebral cortex [2]. After injury, rehabilitation training can effectively reduce damage to brain nerve cells, promote axonal regeneration, and improve the function of the CNS circuit, thereby promoting structural and functional remodeling of the cerebral cortex and subcortical areas.

##### Synaptic Plasticity

The synapse is the structural basis of functional connection and information transmission between neurons. Synaptic plasticity refers to sustained changes in synaptic development, morphology, and information transmission function. Synaptic plasticity is realized by enhancing the signal transmission efficiency of synaptic sites between existing neurons, and it is one of the bases for functional recovery after neural damage. Rehabilitation training is beneficial for synaptic plasticity, and it may enhance the stability of new synapses formed in the first few weeks after cerebral infarction in rats [65].

In adults, voluntary exercise can increase the number of dendritic spines [66], as well as the density and length of dendrites in cognitive-related areas, such as the hippocampal horn and dentate gyrus [67]. SCI affects the resting potential of motor neurons in the CNS, and roller rehabilitation exercise in rats with SCI is beneficial for maintaining the resting potential of motor neurons [68]. Many animal models and studies after spinal cord and cerebral nerve injury support the conclusion that rehabilitation training promotes synaptic plasticity and neurogenesis [69].

The effect of rehabilitation training on synaptic plasticity is mainly influenced by various factors at the molecular level of synaptic structure. The mechanism of synaptic plasticity induced by rehabilitation training involves a large number of molecules involved in the maintenance and regulation of brain function, including neurotrophic factors, signal transduction proteins, transcription factors, and synaptic proteins [70,71].

Firstly, rehabilitation training can promote synaptic plasticity by modulating the molecular composition of presynaptic scaffolds. Scaffold proteins play a crucial role in numerous synaptic functions. The scaffold proteins implicated in synaptic plasticity mainly include GAP-43, synaptophysin, and PSD-95. Growth-associated protein (GAP)-43, a neuron-specific protein, is also closely related to neurogenesis, synaptic plasticity, and regeneration [72]. Mizutani et al. discovered that voluntary exercise leads to an upsurge in GAP-43 levels in the ischemic cortex and improves functional recovery [73]. Synaptophysin, a glycoprotein attached to the synaptic vesicle membrane, performs an important function in the biogenesis of synaptic vesicles [74]. Reduced synaptophysin adversely affects synaptic plasticity in the brain [75]. Motor skill training can heighten synaptophysin expression in stroke rats and promote synaptic regrowth in the thalamus [76]. As a postsynaptic marker, PSD-95 is predominantly found at mature glutamate synapses [77]. The decrease in PSD-95 leads to neuronal loss and synaptic disintegration, thereby impacting the motor, sensory, and cognitive function of the brain [77]. After rehabilitation training, the protein levels of PSD-95 in the sensorimotor cortex noticeably increase [78]. Interestingly, in a study of rats after cerebral ischemic injury, the level of PSD-95 in the bilateral hippocampus in the low-intensity exercise group was higher than that in the high-intensity exercise group [79]. Therefore, low-intensity exercise may be more beneficial for synaptic plasticity than high-intensity exercise after cerebral ischemic injury.

Secondly, rehabilitation training can promote synaptic plasticity by regulating synaptic plasticity regulatory proteins. The upregulated expression of neurotrophic factors and certain synaptic-related proteins performs an essential role in the rehabilitation of cerebral ischemic injury [73]. Rehabilitation training can impact the expression of these proteins and regulate synaptic structure and signal transmission. Common neurotrophic factors such as BDNF, GAP-43, and insulin-like growth factor (IGF-1) contribute to the formation of synaptic plasticity. BDNF phosphorylates its TrkB receptor, controls the actin cytoskeleton in dendritic spines [80] and its degeneration [81], and promotes actin polymerization [82]. BDNF may also play a key role in synaptic plasticity by controlling PSD-95 transport [83]. Rehabilitation training can significantly increase BDNF levels in the brain, thereby enhancing synaptic plasticity [84]. Calmodulin-dependent kinase II (CaMKII) is a rich synaptic signal protein molecule. CaMKII is essential for the long-term enhancement of learning and memory and performs a synaptic role in synaptic plasticity [85]. CaMKII can bind to MAP2 and actin to mediate synaptic transmission [86]. The expression of CaMKII in skeletal muscle increases with endurance training [87] and aerobic training [88]. After 10 weeks of aerobic exercise, the number of synapses and the expression of the synaptic plasticity-related protein CaMKIIα in the hippocampus of mice are noticeably increased, and aerobic exercise can reduce neurodegeneration in the hippocampus by regulating the CaMKII signal transduction cascade [89].

In addition, rehabilitation training can contribute to the formation of synaptic plasticity by modulating the molecular organization of postsynaptic membrane receptors. The AMPA and NMDA receptors are the key mediators of excitatory synaptic transmission in the brain. They are glutamate-gated cation channels that convert chemical signals (glutamate released from presynaptic terminals) into electrical signals (changes in membrane voltage) [90]. AMPA receptors can effectively regulate excitatory neurotransmission and activity-dependent plasticity [91]. The expression of AMPA glutamate receptor subunits (GluR1 and GluR2/3) increases with prolonged exercise time, indicating that the enhanced plasticity of GluR is involved in the process of exercise-induced synaptic plasticity [92]. Exercise also increases the expression of GluA2 subunits in postsynaptic proteins in the cerebral basal ganglia and induces changes in the expression of GluA1 and GluA2 subunits of AMPAR [93]. In the subacute stage of stroke, exercise intervention can significantly increase the mRNA levels of GluA1 and GluA4 and improve synaptic transmission and brain plasticity [91]. NMDA receptor is an ionic glutamate-gated receptor, which is composed of NMDA receptor subunits GluN1, GluN2, and GluN3. The composition of NMDA receptors is strictly regulated by activity-dependent synaptic plasticity during development. Treadmill exercise can improve the expression of NMDA receptors in rats [94]. Treadmill exercise inhibits neuronal apoptosis by increasing the expression of NMDA receptors [95].

It should be noted that more synapses do not always mean better brain function. A published study of brain tissue suggests that children with autism have a surplus of syn-apses [96]. Nagata et al. [97] also found that in mouse models of schizophrenia, extra-large synapses are over-represented, which evoked supra-linear dendritic and somatic integration, resulting in increased neuronal firing. The probability of extra-large spines correlated negatively with working memory [97].

##### Effect of Rehabilitation Training on the Structural Remodeling of the Cerebral Cortex

Following SCI, increased sensitivity of peripheral receptors and neuroinflammation lead to the apoptosis of cortical neurons, and a decrease in the number of cells results in a decrease in the volume of the cerebral cortex [98]. Using fMRI and voxel-based morphometry, Jurkiewicz et al. [99] and Wrigley et al. [100] found that the gray matter volume of the somatosensory cortex and motor cortex in patients with SCI was significantly lower than that in healthy subjects. Freund et al. [101] also confirmed the presence of structural atrophy of the sensory and motor cortex in patients with chronic cervical SCI, and this condition is closely related to the degree of dysfunction and recovery. The deterioration of cortical structural integrity reduces the processing speed and functional integration of the brain, leading to the deterioration of walking function and neuropathic pain after SCI [6,102].

Cerebral cortical plasticity refers to the ability of the CNS to undergo changes in its existing cortical structure and function in response to learning, training, or CNS injury [103]. Synaptic plasticity is one of the main components of cerebral cortex remodeling. Exercise strengthens synaptic connections and can promote synaptic formation or maintain degenerative synapses after injury [4]. The study of synaptic and axon regeneration after peripheral nerve injury also supports the view that the structure of the cerebral cortex is malleable [104]. Rehabilitation training is an important means to promote the structural reorganization of the cerebral cortex after SCI. The improvement of function after exercise therapy is related to the degree of activation of the motor cortex [13]. Studies on the structural remodeling of the cerebral cortex have shown that treadmill training can promote axonal growth [68], sprouting of diseased proximal lateral branches, and synaptic establishment [105]. The activation of dormant axons, which protrude axonal buds and grow new lateral connections, may increase the volume of the cerebral cortex and the structural remodeling of the cerebral cortex [7]. In neonatal rats, treadmill training induced a new type of tissue in both the somatosensory cortex and the motor cortex, which is directly related to the number of weight-bearing steps that animals can take when exercising on the treadmill [106,107]. Exercise can also have a neuroprotective effect on the brains of adult rats [108].

##### Effect of Rehabilitation Training on Functional Remodeling of the Cerebral Cortex

The spinal cord and brain have a close functional relationship. SCI affects the functional network of the motor sensory brain area and the functional connection between the cortexes [109]. Kaushal et al. [110] tested the resting-state functional connections (rs-FC) of the whole brain in patients with complete SCI and found that compared with the control group, their connectivity in the brain functional network decreased, whereas the rs-FC between the cerebellum and bilateral paracenter lobules increased. Hou et al. [111] first analyzed the whole brain ALFF of 25 patients with SCI. The results showed that FC increased in the motor network in the cerebral hemispheres, while FC in the primary motor cortex between the bilateral hemispheres decreased.

Rehabilitation training affects the structure of the cerebral cortex, as well as the function of specific brain regions or functional networks across brain regions. Considering that the motor cortex is one of the main areas that dominate movement, rehabilitation training is usually closely related to the plasticity of the motor cortex [102]. A longitudinal fMRI study of patients with cervical SCI showed that the improvement of function after exercise therapy was significantly correlated with the degree of activation of the motor cortex [13]. Several SCI rehabilitation training studies have shown that the above correlation is widespread in clinical patients [112].

In addition to changes in cortical functional activity in specific areas, functional compensation of neuronal populations in adjacent cortical areas is also a characteristic of cerebral cortical functional remodeling. Rehabilitation training after SCI can expand the activation range of the cortical area and promote the compensation of the participating function of the adjacent cortex of the denervated brain area, and this finding has been confirmed by electroencephalogram (EEG) [113], fMRI [114], and positron emission tomography studies [115]. Considering the spatial distance, the compensation of participating functions in the adjacent cortex and co-activation of neurons in adjacent areas may result from the establishment of synaptic connections induced by local rehabilitation training [116].

The functional network connections across brain regions are the basis for integrating, processing, and controlling sensorimotor information [117]. Rehabilitation exercise also regulates long-distance functional connections across brain regions, such as inter-hemispheric connections. Sawada et al. [118] conducted dexterous finger rehabilitation training in rhesus monkeys after SCI and used cortical electroencephalography to reveal the interaction between cortical activation and behavior. The results showed that grasping training promoted the recruitment of related networks in the contralateral hemisphere of rhesus monkeys from PM to M1, as well as the recruitment of related networks between the contralateral PM of SCI and the ipsilateral PM of SCI. The interaction between the cerebral hemispheres provides a compensatory mechanism that facilitates the activation and interaction of neurons in the bilateral motor cortex to promote functional recovery. Chisholm et al. [119] conducted motor resistance training in patients with incomplete SCI. The results showed that the sensory excitability of the body and corticospinal excitability were enhanced, and the FC value of one limb was increased with less functional impairment. This finding shows that functional training causes brain function reorganization in a short time, which is of great significance to its functional recovery. Studies on brain diseases such as stroke have further demonstrated the regulatory effect of rehabilitation exercise on the reorganization of brain functional networks. In patients who experience stroke, the unilateral voluntary handshake task can increase the activation of the M1 cortex on the affected side [120]. At the same time, early high-intensity upper limb training can increase the activation of the ipsilateral premotor area and anterior cingulate cortex and reduce the activation of the contralateral cerebellum [121]. The voluntary unilateral rehabilitation task enhanced the excitatory connection from the affected side to the contralateral M1 cortex [120]. These clinical studies on stroke have confirmed that exercise training can promote functional reorganization between cerebral hemispheres and improve motor ability by increasing the activation of the ipsilateral brain and enhancing the excitatory connection from the ipsilateral to the contralateral M1 cortex.

##### Maladaptive Plasticity in the Brain

Plasticity refers to the shaping and re-shaping of neural networks at the global level, as well as the remodeling of synaptic contacts at the local level. The plastic reorganization of the brain after CNS lesions plays a crucial role in the recovery and rehabilitation of sensory and motor dysfunction, but it can also be maladaptive [112]. In fact, cortical reorganization is neither beneficial nor harmful: the positive aspect of cortical reorganization can facilitate functional recovery [122,123], while its negative aspect can be maladaptive and lead to phantom sensation and neuropathic pain [112,124,125].

It has been confirmed that brain plasticity responses may result in maladaptive plasticity. Jiang et al. [126] found that reorganization in the primary somatosensory cortex of rats and mice with SCI leads to neuropathic pain. Beeler et al. [127] found that aberrant cortical learning in Parkinson’s disease patients impairs cortical compensation mechanisms, exacerbating functional decline. Nudo et al. [128] also found that in the squirrel monkey stroke model, more proximal hand representation areas spread into the former finger representation area but are negatively associated with the recovery of hand function after the lesion. Therefore, a more comprehensive understanding of the effects of plasticity and brain reorganization is necessary.

#### 2.2.5. Summary of This Section

We have briefly evaluated the potential mechanisms of exercise that affect the brain structure and function, as well as the rehabilitation training process for the brain after SCI (Figure 1). Understanding the regulatory role of rehabilitation training at the supraspinal center is of great significance for clinicians to develop SCI treatment strategies and optimize rehabilitation plans.

## 3. Different Rehabilitation Techniques and Their Effects on Brain Reorganization after SCI

### 3.1. Rehabilitation Exercise Regulates the Brain

Rehabilitation-exercise-induced cortical reorganization mainly depends on the principle of motor learning. Motor learning has benefits for dendritic sprouting, formation of new synapses, modulation of existing synapses, and production of neurochemicals [129]. Sports learning produces remarkable results when carrying out meaningful, repetitive, and intensive rehabilitation exercises [130]. Functional recovery through motor learning can be classified into two types: substantive and compensatory motor recovery. Substantive motor recovery occurs in an uninjured or alternative pathway and transmits nerve impulses to the same muscles used before injury, while compensatory motor recovery uses other muscles different from the original muscles to achieve the same goal [131]. For patients with SCI, substantive motor recovery is the goal of functional recovery; however, it is often difficult to achieve due to damage to conduction pathways and the lack of spontaneous regeneration. Compensatory motor recovery can be achieved by means of residual neural pathways and brain central reorganization, and it is the most important type of functional recovery currently [132].

Many different rehabilitation exercises can be performed after SCI. In animal research, rehabilitation exercises can be divided into voluntary and forced exercises [133]. The most typical voluntary exercise is voluntary wheel running, where animals living alone voluntarily use running wheels for exercise. Forced exercise includes running on a treadmill and swimming in a closed swimming pool. To some extent, these two models are similar to the human rehabilitation exercise model, and they are both beneficial for regulating metabolic protein [134] and BDNF [135] and improving the degree of nervous system injury [133]. Although forced exercise can effectively quantify and reduce isolation stress, some reports suggest that voluntary exercise is beneficial for SCI rats [133] and mice [136].

For human beings, rehabilitation exercises can be classified into anaerobic and aerobic exercise, both of which can promote the change and recovery of brain structure and function after SCI. Chisholm et al. [119] found that anaerobic exercise such as resistance training can improve somatosensory sensation and corticospinal excitability in patients with motor incomplete SCI and regulate brain functional connections at rest. After resistance training, somatosensory excitability and corticospinal excitability were enhanced, and motor-evoked potentials increased [119]. The analysis of fMRI in the resting state of the patients showed that the functional connection of the motor cortex was more obvious in the less affected side after exercise. Aerobic exercise mainly refers to dynamic exercise that relies on aerobic metabolism to promote muscle contraction and the exercise of large muscle groups, including jogging, cycling, and swimming. Aerobic exercise can improve walking and balance function in patients with SCI [137]. It can also improve cerebral blood flow and neurovascular coupling to promote changes in brain structure and function [138]. In addition to SCI, aerobic and anaerobic exercise play an important role in the treatment of brain injury caused by other brain diseases. In individuals who experienced stroke, compared with the impedance exercise group, fixed bicycle aerobic training can improve the speed of information processing [139]. Although both aerobic and anaerobic exercise can independently improve the structure and function of the brain, the effects of the two forms of exercise are distinct. Combining the two modes of exercise is beneficial for improving cognitive function in the brain [139]. During rehabilitation exercises, individuals with severe motor dysfunction generally require one-to-one guidance from rehabilitation therapists to move their paralyzed limbs. This assistance promotes the recovery of muscle strength and the rebuilding of motor nerves [140]. At the same time, various instruments are necessary to support patients during rehabilitation exercises, such as weight-support treadmill training for individuals with SCI. This method reduces the biomechanical and balance constraints of walking by providing weight support through harness systems, allowing for a more normal walking pattern [141].

It is worth noting that exercise may have a negative impact. Chapman et al. [142] discovered that vigorous exercise was connected to a higher risk of amyotrophic lateral sclerosis (ALS). A 2021 study looked at roughly 10% of patients with ALS and found that in these individuals, higher rates of exercise were associated with an earlier onset of the disease [143]. At the same time, some researchers believe that the positive effects of exercise on brain function can be described by an inverted U-shaped curve, suggesting that the appropriate amount of rehabilitation exercise may result in the best possible outcome for the human body [144,145,146]. However, further evidence is needed to explore the potential mechanism and effect of this theory. 

### 3.2. Epidural Electrical Stimulation Regulates the Brain

For patients with severe SCI and those who are unable to exercise autonomously, additional auxiliary means are often necessary to strengthen limb rehabilitation training, and functional electrical stimulation is proposed. Functional electrical stimulation activates muscle contraction by stimulating currents on human muscles, allowing patients to complete the corresponding movements on their own [147]. Epidural electrical stimulation (EES), which is developed from functional electrical stimulation, is a kind of rehabilitation stimulation after SCI with clinical potential. Its mode of action involves the implantation of an electrode into the epidural space of the spinal canal to make electrical stimulation act on the spinal cord tissue. EES transmits a sequence pulse current through the electrode to induce the depolarization of nerve cells in the stimulated part of the spinal cord, causing the dominant muscle to contract to drive joint movement, thereby improving the motor ability of the limbs of patients with SCI [148,149].

The use of epidural stimulation in combination with rehabilitation exercise after SCI can significantly enhance the functional recovery of patients. The first research on the use of EES technology to assist motor function recovery in patients with SCI began in the early 20th century, in which the mode of open-loop stimulation was applied [150]. When performing open-loop stimulation, the stimulator uses a constant or cyclic stimulation mode regardless of the real-time position of the limb and feedback from the brain or peripheral nervous system. This study proved that the combination of EES with exercise training can improve the motor function of patients with incomplete SCI. Since then, researchers have been exploring the closed-loop stimulation model of EES. In the closed-loop mode, EES stimulation can be triggered based on brain activity [151,152,153] or movement at different stages of the gait cycle [154,155]. Closed-loop stimulation aims to activate the target muscle while maximizing sensory inputs [156]. Wagner et al. [157] showed that after a period of closed-loop EES stimulation combined with lower limb rehabilitation training, two patients with SCI were able to autonomously control the paralyzed legs and restore certain functional activities without active stimulation. EES stimulation alone can also promote the recovery of motor and autonomic nervous function in patients with SCI [158]. In another clinical trial, seven patients were treated with EES for 5–9 months without intensive exercise, and more than half of them were able to resume autonomous exercise continuously without active stimulation after EES treatment [159]. The above studies show that epidural electrical stimulation can induce limb movement and proprioceptive sensory nerve input, which can promote the recovery of motor sensory function below the injury level and strengthen the autonomic nervous function of patients.

Electrical stimulation at the site of SCI can also regulate the pathway above the injury and activate the motor cortex [160]. Some studies have confirmed that in the mouse SCI model, EES combined with photogenetic stimulation of the motor cortex can immediately restore weight-bearing movement in mice. When the motor cortical stimulation stops, motor function is also blocked, indicating that EES induces cortical activity to control movement [161]. EES can also increase the level of BDNF in patients [162] and inhibit nerve inflammation at the injured site [163]. Although current neuroimaging studies on the effects of EES on brain structure and function in patients with SCI are not perfect, the combination of EES with repeated rehabilitation exercises plays an important role in the recovery of motor sensation after SCI and activates the ascending pathway of the spinal cord and motor cortex.

During the rehabilitation training process, clinicians need to choose the appropriate EES intensity and mode based on the patient’s condition. The implanted electrode of EES not only has risks related to surgical damage but also needs to maintain biocompatibility for a long time. Hence, the clinical application of this rehabilitation technique requires further research.

### 3.3. Exoskeleton Rehabilitation Robot Regulates Brain

An exoskeleton rehabilitation robot is a mechanical device that simulates the structural characteristics of the human body and attaches to the exterior of the patient’s body to assist or replace the patient’s muscle strength and skeletal joint function, thereby helping patients recover or improve the function of their limbs [164]. Mechanical exoskeletons provide rehabilitation training for patients with CNS injury through their mechanical structure and control system [165], and this technique has the characteristics of high efficiency, quantification, and real-time interaction [166].

Currently, the main training modes of wearable upper limb exoskeleton rehabilitation robots include the passive, cooperative, and active modes. In the passive mode, the patient is in a completely relaxed state, and the exoskeleton drives all parts of the upper limb to the intended target position. Most mechanical exoskeletons are equipped with this mode. The cooperative mode improves upon the traditional passive mode, and the patient can drive the exoskeleton according to the trajectory of movement. When the collaboration mode is used, the exoskeleton can judge whether the exoskeleton is helpful or not based on the motion state of the patient. Patients actively participate to some extent, which plays a better role in promoting the reconstruction of the nervous system and the recovery of motor function [167]. Through rehabilitation-robot-assisted training, patients’ walking ability [168], walking speed [169,170], leg muscle strength [171], stride length [172], and gait symmetry [173] have been improved.

Rehabilitation-robot-assisted training is widely used in artificial therapy after SCI, thereby providing early, intensive, task-specific motor sensory training [174], and it can also promote the adaptive plasticity of brain and spinal cord sensorimotor networks [175]. Chintan and Simon [176] compared rats that received only running training with those that received active stepping robot-assisted running training after SCI. The results showed that the motor characteristics of the cortex of rats with robot-assisted rehabilitation training were abundant, and the motor cortex was effectively activated. Effective robot rehabilitation training can induce the reorganization of the motor cortex and partially reverse some plasticity changes. Chintan and Simon [176] used functional near-infrared spectroscopy technology, which utilizes the scattering properties of blood’s main components to near-infrared light to obtain the changes in oxygenated hemoglobin and deoxyhemoglobin during brain activity, to explore the role of robot-assisted rehabilitation in the rehabilitation of patients with SCI. The results showed that the activity of the motor cortex increased significantly during treatment. In addition to research on patients with SCI, the effect of rehabilitation robot-assisted training on cerebral cortex reorganization has also been examined. In a clinical study of rehabilitation robot-assisted gait training for 40 patients who experienced stroke, the researchers observed an improvement in the effective connection between the frontal and parietal cortex after eight weeks of Ekso^TM^ rehabilitation robot training compared with regular training [177]. Wagner et al. [178] studied the spectral patterns of active and passive robot-assisted walking through EEG signals. The results indicate that both active and passive robot-assisted training can stimulate the cerebral cortex. Furthermore, during active robot-assisted training, changes in cerebral cortex activation are associated with the gait cycle stage. 

The positive role of exoskeleton rehabilitation robots in assisting rehabilitation training has been preliminarily confirmed. In the future, the rehabilitation mechanical exoskeleton will need to further effectively exchange dynamic information with patients to achieve man–machine fusion. This type of rehabilitation training can effectively improve the active participation of patients and significantly improve the rehabilitation effect.

### 3.4. Motor Imagination Rehabilitation Regulates the Brain

In addition to rehabilitation methods that indirectly stimulate the CNS of the brain by passively acting on the distal limbs, new rehabilitation methods have been developed to directly reshape the cerebral cortex by making full use of the subjective intention of patients in order to improve the efficiency of rehabilitation. Among these methods, motor imagination (also known as psychological imagination) refers to the execution of specific actions or tasks without actual signal output [179]. Motor imagination (MI) can activate areas in the primary motor cortex, cerebellum, and basal ganglion circuits [180]. It can also induce functional redistribution and regulation of neural circuits, and then reshape the brain neural network and improve the relearning ability of motor function [181]. In the early stages of SCI, most of the structures and functions of the brain are preserved, providing a matrix for motor imagination [182]. MI therapy does not rely on the residual function of patients, which can give full play to the subjective initiative of patients with SCI and can run through the whole process of rehabilitation of patients with SCI, making it important for the functional recovery of patients with SCI [183].

Sabbah et al. [184] conducted MI training in patients with complete SCI and healthy individuals. Based on the randomized controlled trial, the activated brain regions of the two groups were generally similar during motor imagination, and the primary motor cortex, supplementary motor area, and premotor cortex were all activated, demonstrating that patients with SCI could carry out motor imagination training. Chen et al. [185] performed the right ankle dorsiflexion metatarsal flexion, motor imagination task, and motor execution task in 17 patients with incomplete SCI with partial motor dysfunction of the right ankle. The results showed that the activated brain areas were generally similar, including the bilateral supplementary motor areas and inferior frontal gyrus. However, compared with the motor execution task, the activation degree of some areas in the motor imagination task was lower. Rienzo et al. [186] found that after adding MI training to the rehabilitation program of patients with SCI, the activation of compensatory brain areas decreased, and the cortical activity was closer to the normal state, suggesting that MI training partially reversed the compensatory neural activation in patients with SCI, which was beneficial to the integration of normal neural networks. The latest study by Wang et al. [187] compared the effect of MI in children with SCI with that in healthy subjects by fMRI. The results showed that the activation degrees of bilateral paracentral lobules, auxiliary motor area, putamen, and cerebellar lobules in patients with SCI were higher than those in healthy ones.

The above studies indicate that motor imagination enhances the motor rehabilitation of patients with SCI. MI can also be combined with brain–computer interface technology, which can restore part of the patient’s motor function without relying on the normal nerve conduction pathway of the brain [188]. Motor imagination–brain–computer interface will be an important future direction for the recovery of motor function after SCI. Current research on the effects of MI mainly shows that motor imagination can improve the CNS function of patients with brain injury in the short term, but its long-term effects on brain structure and function reorganization require further investigation.

### 3.5. Comparison of the Advantages and Disadvantages of Different Rehabilitation Methods

Different rehabilitation training methods have varied characteristics and effects (Table 1), and their effects on brain structure and function also differ. In clinics, the most suitable rehabilitation program is typically chosen based on the patient’s status in order to achieve the best possible outcome [189].

## 4. Summary and Prospects

SCI and rehabilitation training both have the ability to modify the structure and function of the brain. This paper provides an overview of the physiological mechanisms responsible for the effects of rehabilitation training on brain structure and function after SCI and introduces and compares the effects of rehabilitation exercise, epidural electrical stimulation, mechanical exoskeleton, and motor imagination on the activation and reorganization of the cerebral cortex. Although rehabilitation training has been widely used in the clinical treatment of SCI, its impact on promoting brain remodeling to enhance motor function remains to be quantitatively evaluated. Understanding the influence mode and law of rehabilitation training on changes in brain structure and function, as well as the regulation and degree of brain remodeling on motor function after SCI, will help guide the clinical development of personalized rehabilitation strategies. These findings have important implications for the treatment of SCI. 

## Figures and Tables

**Figure 1 biomedicines-12-00041-f001:**
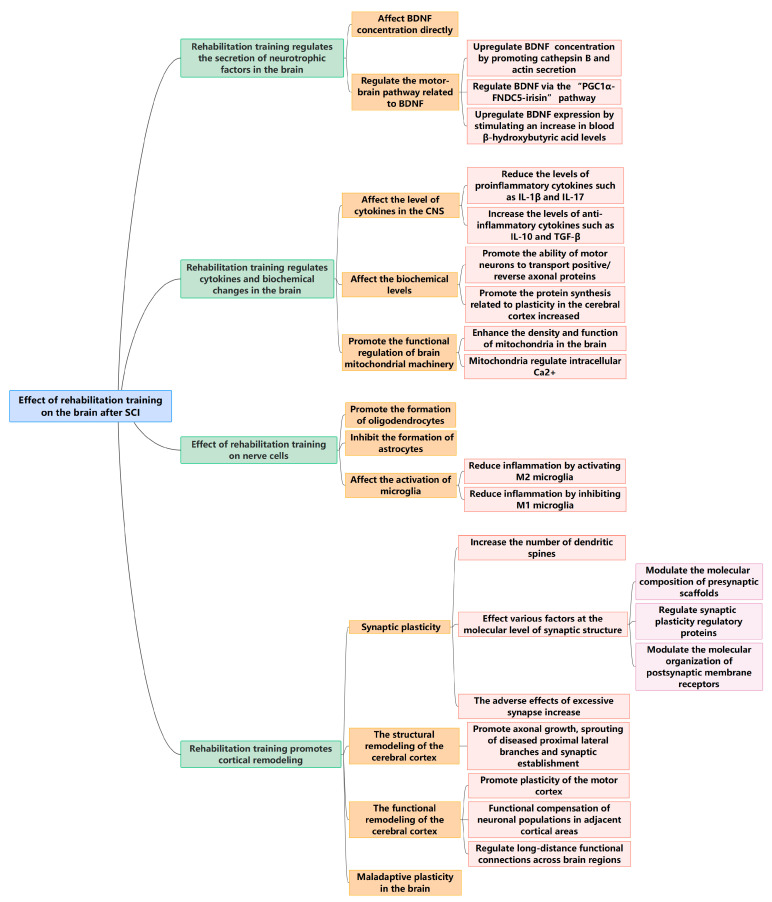
Effect of rehabilitation training on the brain after SCI.

**Table 1 biomedicines-12-00041-t001:** Characteristics and effects of different rehabilitation training methods.

Types	Characteristics	Advantage	Shortcoming	Effect	Applicability
Rehabilitation exercise	Easy to operate, convenient, and low-cost; the most commonly used clinical method for the rehabilitation of spinal cord injury (SCI).	Practicability, operability, and low cost; the most effective rehabilitation means to promote brain reorganization after SCI at present.	Exercise acts on the distal limbs, indirectly stimulates the CNS of the brain; rehabilitation training takes a long time.	One of the most important quantifiable methods of functional recovery after SCI, which activates the cerebral cortex by acting on the distal limb.	It can be applied to people with various degrees of SCI and at various stages of treatment after SCI, but it is mainly used for incomplete SCI.
Epidural electrical stimulation (EES)	It can induce movement through a sequence of pulsed currents and makes effective use of the residual function of spinal nerves and muscles.	It can enhance the neural plasticity of the motor cortex and activate the excitability of the corticospinal tract.	The use of implantable electrodes may cause postoperative infection. Prolonged application of electrical stimulation can cause pain in patients.	Combined with rehabilitation exercise, patients can improve their exercise ability in a short period of time.	EES is suitable for severe and complete SCI. It can be used for the early rehabilitation of SCI and patients with severe motor dysfunction.
Exoskeleton rehabilitation robot	It can simulate the structure characteristics of the human body and has the characteristics of high strength, repeatability, and interaction.	It can provide repeatable and specific movement therapy and promote the rehabilitation of fine movements after SCI. It can objectively quantify rehabilitation parameters and training output.	The structure is complex and expensive, and the weight and body shape should be considered.	It can be used to provide standard movement training for injured limbs. It has a better effect on brain reorganization than routine exercise training.	For patients with severe and complete SCI, it can provide more effective support. It is used for rehabilitation training of patients with severe motor dysfunction after SCI.
Motor imagination (MI)	Makes full use of the patient’s subjective intentions and acts directly on the cerebral cortex.	It has a wide range of applications, motivates patients, and guides patients to achieve appropriate brain activation patterns related to tasks.	The standard of treatment and the best intensity of intervention are not clear. The effect depends on the feedback content and the status of the patient.	It can help patients with SCI activate the activity of the cerebral cortex and realize the self-regulation of the functional brain networks. Combining with other rehabilitation training methods is beneficial to the better effect of MI.	It can be applied to individuals with complete and incomplete SCI and at various stages of treatment after SCI.

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
