# Peer review of "Rehabilitation Training after Spinal Cord Injury Affects Brain Structure and Function: From Mechanisms to Methods"

_biomedicines, 2023, doi:10.3390/biomedicines12010041_

Round 1

Reviewer 1 Report

Comments and Suggestions for Authors

The review is from a group that is and has been active in the field for many years and as contributors, know a wide range of information about SCI, which they carefully lay out in this review. They could add information about proprioception as it would fit in well. Figure 1 though an adequate summary figure is impossible to read and could easily be changed in a way that lettering was larger. Regarding rehabilitation in humans, the review could add a bit more information as to the suitability of different rehabilitations based on the injury, the age and level of recovery. For example, in Table 1, five categories (type, characteristic, advantage, shortcoming and effect) are included, but it is difficult to figure which individuals are matched for different rehabilitations. 

Reviewer 2 Report

Comments and Suggestions for Authors

This is a review on an interesting topic. I have the following suggestions for revision:

Line 50 replace the word “study” with “manuscript”

Line 98 instead of “muscle tubes” did you mean “myotubes?”

Line 101 what is the significance of the increase in actin here?

 Lines 104 to 110. The description of the mechanisms in this section is unclear.  I suggest rewriting this section.

 Line 122: “ make the level of interferon IFN-γ return to normal.” the significance of this change is unclear.

Lines 133 to 134: “ PGC-1a and SIRT1, activation of AMPK, de- 133 crease of p53 acetylation” - again the significance of these changes needs to be explained

Lines 265 to 267 “The ways and effects of rehabilitation training on regulating BDNF secretion have been discussed, and many studies have shown that rehabilitation training is beneficial to the increase of BDNF levels in the brain.” - I think this sentence can be deleted because it is repetitive with the next sentence.

Line 383: I think a “chapter” is not the right word to use here

Line 418: please explain what is meant by “impedance exercise”

Lines 420 to 424: references are needed here to support these statements on the benefits of resistance exercise

Lines 437 to 438: “ This assistance promotes the recovery of muscle strength and the rebuilding of motor nerves.” - again this needs a supporting reference.

Line 441: “seatbelts” is not the appropriate term to use here. I suggest “harness systems.”

Line 442-443: “ The positive effects of exercise on brain structure and function can be described by an inverted U-shaped curve” - this needs to be clarified. I am not sure if this is true.

Lines 493 to 494: I suggest rewording this sentence

Line 527: “ fNIRS” - I suggest spelling out this abbreviation and explaining this measurement technique

Line 534: should TM be superscripted here?

Line 536: the significance of this change is not clear. What does this mean?

Line 560: “MI” - this abbreviation needs to be explained

Line 561: “ control experiment” - is this the correct term to use here?

Line 575: “ normal” - I suggest using a different term such as “healthy” or “in injured”

Line 591: it is unclear what “actual situation” refers to here

Table one: I don’t think you have discussed functional electrical stimulation as a method of rehabilitation in your article. I suggest adding a short section on this.

Comments on the Quality of English Language

The English is fine

Reviewer 3 Report

Comments and Suggestions for Authors

The review article entitled “Rehabilitation training after spinal cord injury affects brain structure and function: from mechanisms to methods” by He and coworkers reflects on the pathological outcomes that spinal cord injury has on the upper and lower levels of the neuraxis, while discussing some exercise-based techniques to improve the quality of life of the SCI-patients. Even though the manuscript is interesting, and welcomed, there is a profound theoretical misunderstanding that must be corrected before being considered seriously for publication. The concept that “When SCI occurs, it cuts off the crucial information flow between the peripheral nerves and the brain, thereby affecting not only the areas below the level of injury but also the structure and function of the brain” is at best misleading. The information flow that SCI limits (below the section’s level) or modifies (above the section level) is not between the nerves and the brain. It is between the whole body and the brain, being the body the upmost important source of conscious and unconscious somatic, vegetative, and trophic information and support for the brain. Such a misconception must be corrected. In doing so, I recommend authors to read Body and Brain: The trophic theory of neural connections by Dale Purves, Harvard University Press, 1988 (available at his web site Purves Lab for free). There is also another notorious epistemic mistake when saying that “The CNS has a remarkable ability to adapt and compensate for damage”. This affirmation suggests to the reader that “brain compensation” has an intention. This is far from truth. Remaining neurons and the damage ones are trying to survive while searching for sources to gain trophic support. They are not, a priori, looking for compensation. Compensation is fully circumstantial. This is supported by the fact that much of the plastic response may lead to maladaptive plasticity, a theme that is barely commented in this review; reviews must be critical, especially because some rehabilitation measures may reinforce maladaptive plasticity. When authors write” When the balance of the brain is disrupted, the cerebral cortex can undergo neural network reorganization to adapt to the changes”. What do they mean with balance? Would not be plasticity outcomes a sort of a new balance?  Does the cerebral cortex look for adaptation? How this comes since adaptation is circumstantial. What is good for somethings, might not be so under a different context. Authors must discuss maladaptive plasticity in their review, otherwise is profoundly skewed. Another misleading concept is assuring that the creation of new synapses is a good sign of a healthy brain or of a brain ongoing adaptation. This notion must be revised. Authors must recall that, for the brain to function well more synapses not always mean, better function. Plenty of examples illustrating my point. In fact, muscle immobilization, increases the number of neuromotor connections, leading to awkward movements. Physical rehabilitation brings the number of neuromuscular contacts down, thus improving motor function. In addition, I would suggest authors not to bring data on the effects of exercise on healthy brains as a good proxy on what could be improved with exercise following spinal cord injury, because it is not.  Another issue that must be commented on and fairly discussed is the negative effects that exercise may induce in the brain. Not all is good. An there is some evidence on this issue in patients with muscular dystrophies.

Finally, I would suggest authors to consider a different strategy to organize the manuscript, since they promise to talk about mechanisms (models of the mechanisms must be disclosed first, paying particular attention to the description of their premises and predictions) and therapeutic methods (methods must be explained based upon the theoretical background set by the presumptive mechanisms):

A first section aimed at introducing Body and Brain interactions and the trophic theory of neural connections would set a nice rational to understand SCI. As second section laying down the clinical description (above and below the section stie) of SCI, and the current model that intends explain such complex clinical scenario, finding a way to tight it with the trophic theory. Since the current model used to explain signs and symptoms after SCI has specific premises and predictions, authors must introduce their therapeutic proposals based on the premises and predictions of the SCI model, such measures are trying to alleviate. The following section must recognize maladaptive plasticity in the context of SCI outcomes and treatment and talk about the limitations of current measures (specifically those exercise based). A final paragraph on future developments and directions will help the reader to see the upcoming future.  

In sum, authors must bring to their review a more critical and sophisticated posture on the topic they address.     

Comments on the Quality of English Language

Could be improved if edited by an native English speaker

Round 2

Reviewer 2 Report

Comments and Suggestions for Authors

The authors have adequately addressed my comments

Comments on the Quality of English Language

The English is acceptable

Reviewer 3 Report

Comments and Suggestions for Authors

I thank authors for their efforts to improve the manuscript. I have no furher concerns.